# The benefits of full data shuffle, now with optimal I/O cost: $k$-wise independence and matrix transposition to the rescue

**Peyman Afshani** [* 1]  **Rezaul Chowdhury** [* 2]  **Mayank Goswami** [* 3]  **Jens Kristian Refsgaard Schou** [* 4]
**Francesco Silvestri** [* 5]  **Mariafiore Tognon** [* 5]

## Abstract

It is known that RANDOMSHUFFLE, the without replacement version of Stochastic Gradient Descent (SGD), converges faster than with replacement SGD. However, RANDOMSHUFFLE requires uniformly performing a random permutation of the input sequence, which is known to have high I/O complexity due to data movement across the memory hierarchy. In this paper, we propose a shuffling algorithm with a linear I/O complexity that generates almost-uniformly random permutations with rigorous mathematical guarantees. Specifically, we show that the shuffling algorithm can generate 2-wise independent permutations. Furthermore, we can extend to $k$-wise independence with a small error in the probability distribution, if the fast memory has at least $k$ memory blocks. These results allow us to reach the same expected theoretical convergence as RANDOMSHUFFLE while achieving optimal linear I/O cost.

## 1. Introduction

For $1 \leq i \leq N$, let $f_i : \mathbb{R}^d \to \mathbb{R}$ be smooth real-valued functions, and define $F(x) = \frac{1}{N} \sum_{i=1}^{N} f_i(x)$. A classical approach to the problem of finding a minimum of $F(x)$ is *Stochastic Gradient Descent (SGD)*, or the Robbins-Monro algorithm (Robbins & Monro, 1951). An exciting line of recent work has provided theoretical justifications for the improved convergence of RANDOMSHUFFLE

over with replacement SGD, including (Haochen & Sra, 2019; Gürbüzbalaban et al., 2021; Yu & Li, 2023; Nagaraj et al., 2019). Following the conventions in these previous works, we first define both methods. Both SGD and RANDOMSHUFFLE use a step size $\gamma$ that is usually decided before the algorithms are run. Starting from an initial point $x_0$, the algorithms are run for $T$ iterations, with the iterates obtained being denoted as $\{x_t\}_{t=1}^{T}$. In SGD, the iterates are generated in the following manner: at each step $1 \leq t \leq T$, pick an index $s(t)$ uniformly at random from $[N]$, and define

$$x_t = x_{t-1} - \gamma \nabla f_{s(t)}(x_{t-1}).$$

For RANDOMSHUFFLE, one instead does the following: let $\ell = T/N$ denote the number of epochs. Every epoch $e$ (for $1 \leq e \leq \ell$) will have $N$ iterates within the epoch, denoted as $x_t^e$ for $1 \leq t \leq N$. Let $x_0^1 = x_0$ be the initial starting point (of epoch 1), and we will set $x_0^{e+1} = x_N^e$ for two consecutive epochs $e$ and $e + 1$. Now, for every epoch $e$, randomly generate a permutation (i.e., shuffle) $\sigma_e : [N] \to [N]$ under the uniform distribution on the space of all $N!$ permutations. The $N$ iterates $\{x_t^e\}_{t=1}^{N}$ within an epoch $e$ follow the rule

$$x_t^e = x_{t-1}^e - \gamma \nabla f_{\sigma_e(t)}(x_{t-1}^e).$$

For strongly convex loss functions, prominent works like (Gürbüzbalaban et al., 2021; Nagaraj et al., 2019; Haochen & Sra, 2019) provide theoretical justifications for the superior performance of RANDOMSHUFFLE over SGD, which has been experimentally observed in (Bottou, 2009). SGD has a well-known $O(1/T)$ convergence guarantee, and in fact, this is known to be tight up to constants, under the strong convexity assumption on $F$. On the other hand, RANDOMSHUFFLE shows a $O(1/T^2)$ convergence guarantee with $\ell = T/N$ epochs. While the original results by (Gürbüzbalaban et al., 2021) can be seen to show that RANDOMSHUFFLE beats SGD asymptotically, the results in (Haochen & Sra, 2019) prove that RANDOMSHUFFLE beats SGD given sufficiently many (finite) epochs, and (Nagaraj et al., 2019) provides slightly better bounds in the low-epoch regime, when $\ell \lesssim N$.

While RANDOMSHUFFLE has superior guarantees with respect to SGD, the running times of both algorithms are

---

[*]Equal contribution [1]Department of Computer Science, Aarhus University, Aarhus, Denmark [2]Department of Computer Science, Stony Brook University, New York, USA [3]Department of Computer Science, Queens College, New York, USA [4]Scientific Computing and Imaging Institute, The University of Utah, Utah, USA [5]Department of Information Engineering, University of Padova, Padua, Italy. Correspondence to: Francesco Silvestri <francesco.silvestri@unipd.it>.

*Proceedings of the 43rd International Conference on Machine Learning*, Seoul, South Korea. PMLR 306, 2026. Copyright 2026 by the author(s).

significantly affected by the cost of reading data. As models are learned on large collections of data, a significant amount of the running time is dedicated to moving data among the memory hierarchy rather than the actual CPU cost of training (Xu et al., 2024). Being aware of data movement costs can significantly reduce time: for example, FLASHATTENTION is an algorithm to compute the attention values that exploits the memory hierarchy and achieves a ×6 increase in throughput compared to the standard PyTorch implementation (Dao et al., 2022; Dao, 2024).

To capture data movements among the memory hierarchy, we use the *I/O model* (Aggarwal & Vitter, 1988): it consists of a fast but limited memory of size $M$ words and a slow but conceptually unlimited main memory. Transfers of data between fast and slow memory is done by reading or writing blocks of $B$ words, which encourages blocked access to data as opposed to random access. The I/O cost of an algorithm is the number of block movements between slow and fast memories, so simply sequentially reading every datapoint requires $O(N/B)$ I/Os. The I/O model has been widely used for analyzing the cost of data movement, such as in the attention computation (Dao et al., 2022) and in linear algebra computations (Ballard et al., 2011). Within the I/O model, Gradient Descent incurs $O(N/B)$ I/Os per epoch, whereas the standard SGD incurs $O(N)$ I/Os, as it randomly selects documents stored in slow memory. On the other hand, generating one uniformly random permutation for RANDOMSHUFFLE requires $\Omega\Big(\min\Big(N, (N/B)\log_{M/B} N/B\}\Big)\Big)$ I/Os (Aggarwal & Vitter, 1988).

Training models without permuting data reduces the I/O cost; however, the quality of the model is significantly affected. Indeed, the large datasets used in training steps are usually stored according to certain indices (e.g., B-trees, K-NN graphs). However, these indices might impose some structure over the data that can affect the convergence and accuracy of training the learning models: models might learn patterns based on the sequence in which data is presented rather than the correct properties (Xu et al., 2024; Haochen & Sra, 2019; Kim et al., 2025). As an example, training the VGG19 and ResNet18 models on the cifar-10 dataset sorted by class labels incurs an accuracy drop of 50% with respect to a random permutation of the dataset (Xu et al., 2024).

Recent works have introduced several practical alternative shuffling approaches to reduce I/O costs (e.g. (Gu et al., 2022; Liu et al., 2023; Zhong et al., 2023)), which do not provide strong guarantees on convergence and on the properties of the permutations. The work (Xu et al., 2024) presents an I/O-aware alternative to SGD, named CORGIPILE, motivated by the observation that large datasets are often stored in clusters (e.g., sorted by label and feature), and that exposing such clusters sequentially to SGD can lead to poor optimization behavior. The CORGIPILE algorithm fills fast memory and uniformly shuffles elements in the cache, seeking to break these harmful clusters while incurring the optimal $O(N/B)$ I/Os per epoch, corresponding to a linear scan of the input. However, this approach does not aim to control the global structure of the resulting permutation: while clustering is mitigated, the random properties of the permutation at the block level are not fully characterized, and the approach has not been proven to achieve strictly better convergence rates than SGD unless the cache contains the entire input.

Therefore, in this paper, we address the following question:

> *Does there exist a data shuffle approach that requires $O(N/B)$ I/Os while obtaining a $O\big(1/T^2\big)$ convergency rate as in RANDOMSHUFFLE?*

**Our contribution.** To tackle the above question, we first study how much randomness is required from data reshuffling in order to recover the optimization benefits of uniform random reshuffling under stringent I/O constraints. While fully uniform permutations are the ideal target, generating them incurs non-trivial I/O overhead. Our key insight is that block-level reorganization via matrix transposition destroys clustered structure far more aggressively than local uniform shuffling, while remaining I/O-optimal.

Our approach measures permutation randomness through the uniformity of its $k$-marginals. Informally, a permutation distribution is said to be $k$-wise independent if the joint distribution of the positions of any $k$ data points matches that of a uniform random permutation, and almost $k$-wise independent if this holds approximately (we refer to Section 2.1 for rigorous definitions). This perspective isolates the low-order marginal properties of uniform reshuffling that are actually used by existing optimization analyses. We then obtain these results:

- **I/O-efficient exact 2-wise permutations.** We introduce the algorithm GEN-2-WISE-IND-PERM that shuffles $N$ elements using $O(N/B)$ I/Os and produces an exact 2-wise independent permutation under the tall-cache assumption (Frigo et al., 2012), i.e. $M = \Omega(B^2)$.

- **I/O-efficient almost $k$-wise permutations.** We then introduce the two-round shuffling algorithm IO SHUFFLE that likewise uses $O(N/B)$ I/Os, under the tall-cache assumption, to produce an almost $k$-wise independent permutation with error $O(k^2 B^2/N)$. For $k \leq \sqrt{N}/B$, this yields almost-$k$-wise independence while remaining I/O-optimal.

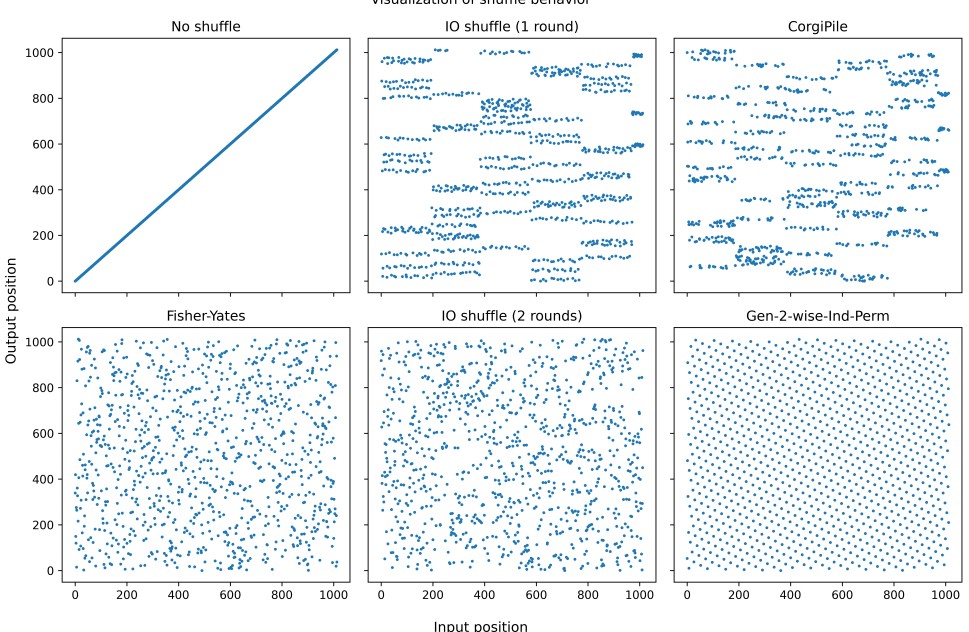

*Figure 1.* Visualization of shuffle algorithms. Each algorithm is executed once on an input with block size $B = 14$ and $N = 1013$ elements. Dots represent (input position, output position) pairs. Visualization inspired by (Xu et al., 2024).

- **Lower bounds.** We show that the tall-cache assumption is necessary: without it, any algorithm producing an almost 2-wise independent permutation must either incur $\Omega(\frac{N}{B} \log_{M/B}(N/B))$ I/Os or suffer a large approximation error.

Figure 1 provides an illustrative comparison of the permutation patterns produced by our shuffling algorithms, a uniformly random Fisher-Yates shuffle, and CORGIPILE (Xu et al., 2024). Note that this visualization is purely illustrative and does not prove uniformity.

Although IO SHUFFLE builds on CORGIPILE, our solution exploits a new approach based on the alternation between matrix transposition and random permutations in fast memory: this approach breaks clustered data faster than the previous method, as experimentally shown in Section 7.

Building on the aforementioned results, we then study the convergence with $k$-wise permutations. We refer to K-WISE-RANDOMSHUFFLE as a variant of RANDOMSHUFFLE where, instead of selecting $\sigma_e$ uniformly at random from the $N!$ possible permutations, one selects it according to a $k$-wise independent distribution. When $k = 2$, we will call the corresponding algorithm 2-WISE-RANDOMSHUFFLE. For the purposes of this paper, we limit our analysis to the strongly convex regime. As mentioned above, for strongly convex loss functions, there are some prominent works giving theoretical guarantees on the phenomenon that RAN-DOMSHUFFLE exhibits a better convergence rate than SGD.

We show in Section 6 the following surprising result when using 2-WISE-RANDOMSHUFFLE:

- All of the results in (Gürbüzbalaban et al., 2021) and (Haochen & Sra, 2019) hold even when RANDOMSHUFFLE is replaced by 2-WISE-RANDOMSHUFFLE.

Therefore, using our Algorithm GEN-2-WISE-IND-PERM implies that, up to a small constant, only a linear number of I/Os are enough to guarantee that 2-WISE-RANDOMSHUFFLE beats SGD in expectation. We also give evidence that K-WISE-RANDOMSHUFFLE obtains the same high-probability guarantees as RANDOMSHUFFLE for strongly convex functions in (Yu & Li, 2023).

The paper is organized as follows: in Section 2, we define the notion of $k$-wise and almost $k$-wiseness and describe related work; in Section 3, we propose and analyze the 2-wise algorithm GEN-2-WISE-IND-PERM; in Section 4, we propose and analyze the almost $k$-wise algorithm IO SHUFFLE; in Section 5, we show the necessity of the tall-cache assumption; in Section 6 we theoretically analyze the convergency of SGD when using with $k$-wise permutations; in Section 7, we experimentally show the efficiency of our solutions in breaking cluster structure; we conclude in Section 8 with some final remarks. Missing proofs are provided in the appendices.

## 2. Preliminaries

In this section, we first define the properties used to measure the randomness of generated permutations. Then we provide an overview of related works on generating permutations and on Gradient Descent.

### 2.1. $k$-wise independence

We study distributions over permutations through the uniformity of their low-order marginals. Let $\mathcal{C}$ be a set of $N$ elements and let $\mathscr{P}$ denote the set of all permutations of $\mathcal{C}$. We view a permutation $\pi \in \mathscr{P}$ as assigning each element $a \in \mathcal{C}$ a unique position $\pi(a) \in [N]$. For an ordered $k$-tuple $S = \langle a_1, \ldots, a_k \rangle$ of distinct elements of $\mathcal{C}$, define

$$\pi(S) = \langle \pi(a_1), \ldots, \pi(a_k) \rangle,$$

and let $\mathcal{I}_k$ denote the set of all ordered $k$-tuples of distinct positions in $[N]$.

Any distribution $\mu$ over $\mathscr{P}$ induces, for each such $S$, a marginal distribution $\mu_S$ over $\mathcal{I}_k$, defined by $\mu_S(I) = \Pr_{\pi \sim \mu}[\pi(S) = I]$. Under the uniform distribution $\mathfrak{U}$ over permutations, the induced marginals $\mathfrak{U}_S$ are uniform over $\mathcal{I}_k$, with

$$\mathfrak{U}_S(I) = \frac{1}{N(N-1) \cdots (N-k+1)}.$$

**Definition 2.1** ($k$-wise independence). *A distribution $\mu$ over $\mathscr{P}$ is $k$-wise independent if, for every ordered $k$-tuple $S$ of distinct elements of $\mathcal{C}$, the marginal $\mu_S$ is uniform over $\mathcal{I}_k$.*

The next definition allows some deviation from the uniform distribution, and the error is given by the $\ell_1$ distance from the two distributions.

**Definition 2.2** (Almost $k$-wise independence). *A distribution $\mu$ over $\mathscr{P}$ is almost $k$-wise independent with total error $\epsilon$ if, for every ordered $k$-tuple $S$ of distinct elements of $\mathcal{C}$,*

$$\sum_{I \in \mathcal{I}_k} \left| \mu_S(I) - \mathfrak{U}_S(I) \right| \leq \epsilon.$$

### 2.2. Related Work

**Permutations** When it comes to generating a random permutation, a common strategy is to use the *Fisher-Yates shuffle* (Fisher & Yates, 1938), which entails operating on an array $A$ of size $N$ via a `for` loop over all the $N$ positions of $A$: at the $i$-th step, the $i$-th element of $A$ is swapped with a random element between $i$ and $N$ (inclusive). It requires $\Theta(N \log N)$ random bits, $\Theta(N)$ time, and $\Theta(N)$ I/Os in the I/O model, which is far away from the natural $O(N/B)$ baseline of linear scanning. In the I/O model it is also possible to generate a random permutation via sorting by first generating a random label for each item and then

sorting the items by the labels, which would however require $O\big((N/B) \log_{\frac{M}{B}}(N/B)\big)$ I/Os and $O(N \log N)$ time, which is sub-optimal. Interestingly, in the I/O model, there are lower bounds that show generating a random permutation requires $\Omega\big(\big(\min\big\{N, \frac{N}{B} \log_{\frac{M}{B}}(N/B)\big\}\big)\big)$ I/Os (Aggarwal & Vitter, 1988). The I/O cost of rational permutations has been analyzed in (Silvestri, 2008), which gives a lower bound matching the sorting I/O cost in the worst case.

Both the problem of generating fully independent random permutations in several models of computation (e.g., see (Bacher et al., 2018; Czumaj et al., 1998; Gustedt, 2008; Penschuck, 2023; Sanders, 1998)) and that of generating less restrictive permutations (such as pseudorandom, $k$-wise or almost $k$-wise independent permutations (Luby & Rackoff, 1988; Alon & Lovett, 2012)) have been abundantly studied in the past. However, to the best of our knowledge, there are no previous works investigating the generation of relaxed notions of permutations in the realistic I/O model.

Random permutations have historically been employed to circumvent worst-case inputs in a number of fields: in computational geometry, common deterministic algorithms (see e.g. (Clarkson & Shor, 1989; Seidel, 1991; Sharir & Welzl, 1992; Matousek et al., 1996; Clarkson, 1995; Gärtner et al., 2008)) exploit a random permutation of the input to achieve good performance in expectation, while permuting adversarial inputs has been used in the random-order model to study algorithms in the streaming setting (Braverman et al., 2018; Gupta & Singla, 2021). Finally, random permutations have been used for discovering causality or extracting statistically significant patterns, for instance, with permutation tests (see e.g., (Dong et al., 2025; Pellegrina & Vandin, 2018)).

**Gradient Descent.** Randomly reshuffling the training data (i.e., the RANDOMSHUFFLE algorithm) has emerged as an alternative to standard SGD, which samples data points with replacement at each epoch. As mentioned above, the superiority of this approach was first observed in practical tasks (e.g. (Bottou, 2009; Recht & Ré, 2013)), but was later theoretically justified, with works such as (Gürbüzbalaban et al., 2021; Haochen & Sra, 2019; Nagaraj et al., 2019) demonstrating its faster convergence rate compared to SGD.

We refer to Table 1 for a summary of the main related works. Let $\kappa = L/\mu$, where $L$ is the Lipschitz constant and $\mu$ is the strong-convexity parameter in (see Appendix C for the definitions). While the original results by (Gürbüzbalaban et al., 2021) can be seen to show that RANDOMSHUFFLE beats SGD asymptotically, the results of (Nagaraj et al., 2019) complement those by (Haochen & Sra, 2019) in the sense that (Haochen & Sra, 2019) proves that RANDOMSHUFFLE beats SGD given sufficiently many (finite) epochs, and (Nagaraj et al., 2019) provides slightly better bounds in the low-epoch regime, when $\ell \lesssim N$.

| Reference | Guarantee | Step Size |
|---|---|---|
| (Gürbüzbalaban et al., 2021) | $\frac{C(N,d)}{\ell^2}$ | $\frac{1}{\ell}$ |
| (Haochen & Sra, 2019) | $O\left(\frac{1}{N^2\ell^2} + \frac{1}{\ell^3}\right)$ | $\frac{\log N\ell}{\mu N\ell}$ |
| (Nagaraj et al., 2019) | $\widetilde{O}\left(\frac{1}{N\ell^2}\right)$ | $\frac{\log N\ell}{\mu N\ell}$ |

*Table 1.* Summary of previous works on RANDOMSHUFFLE gurantees.

---

**Algorithm 1** Gen-2-wise-Ind-Perm($\mathcal{C} = a_0 \ldots a_{N-1}$)

1: Allocate an array $O$ of size $N$, initially empty.
2: Choose $a$ uniformly at random from $[1, N-1]$
3: Let $a^{-1} \in [N] : aa^{-1} = 1 \bmod N$
4: Choose $b$ uniformly at random from $[0, N-1]$
5: Compute $\mathcal{Q} = \{a^{-1}b_1 + b_2 \mid -B < b_1, b_2 < B\}$
6: Compute $\bar{\mathcal{Q}} = \{b_1 + ab_2 \mid -B < b_1, b_2 < B\}$
7: **for** $j = 0$ **to** $N - 1$ **do**
8:    **if** $j$-th entry of $O$ is empty **then**
9:       Let $i = a^{-1}(j - b) \bmod N$
10:      Let $\mathcal{Q}_i$ be the elements at indices $i + \mathcal{Q}$ from $C$
11:      Write $Q_i$ to the indices $ia + b + \bar{\mathcal{Q}} = j + \bar{\mathcal{Q}}$ in $O$
12:    **end if**
13: **end for**
14: **return** $O$

---

Works (Gürbüzbalaban et al., 2021) and (Haochen & Sra, 2019) assume that the number of epochs $\ell$ (equal to $T/N$, where $T$ is the total number of iterates and $N$ the input size) is $\Omega(\kappa^{1.5}\sqrt{N})$, whereas (Nagaraj et al., 2019) and assumes that $\ell$ is $\Omega(\kappa^2 \log(KN))$. Furthermore, in (Yu & Li, 2023), the authors provide a high probability guarantee under the additional assumption that for all $i \in [N]$, each function $f_i$ is lower bounded by $\bar{f}_i$. Their result states that when $T = O(\max\{\sqrt{N}\epsilon^{-3}, N\epsilon^{-2}\} \log(\sqrt{N}\epsilon^{-3}/\delta))$, $\frac{1}{\ell}\sum_{i=1}^{\ell} \|\nabla f(x_i^0)\|^2 \leq \epsilon^2$ with probability at least $1 - \delta$. Finally, we would like to remark that there are more recent exciting results in the non-strongly convex and non-convex regimes in (Mishchenko et al., 2020; Nguyen et al., 2021).

## 3. I/O-efficient exact 2-wise permutations

Here, we study the problem of creating 2-wise permutations in the I/O model under the tall-cache assumption, i.e., when $M = \Omega(B^2)$. Later, in Section 5, we show that the tall-cache assumption is needed to get a linear number of I/Os.

The main challenge that we face here is designing an I/O-efficient algorithm since the approach is very classical: we assume $N$ is a prime number[1] and the idea is to map the $i$-th element to the position $f_{a,b}(i) = ai + b \bmod N$ where

---

[1] If $N$ is not prime it suffices to increase the input size up to the first prime number with a negligible asymptotic cost due to the Bertrand's Postulate (i.e., there exists a prime $P$ in $N \leq P < 2N$).

$a$ is randomly sampled from 1 to $N - 1$ and $b$ is randomly sampled from 0 to $N - 1$ (Carter & Wegman, 1979). This algorithm can be trivially implemented with a simple `for` loop but then it can perform too many random accesses, resulting in $\Omega(N)$ I/Os, in which case we might as well use a fully random shuffling algorithm, such as the Fisher-Yates shuffle. Thus, our goal is to introduce an algorithm that fetches elements that both start together and end together in only $O(N/B)$ I/Os.

The idea behind Algorithm 1 is the following. We define two "squares" $\mathcal{Q} := \{a^{-1}b_1 + b_2 \bmod N \mid -B < b_1, b_2 < B\}$ and $\bar{\mathcal{Q}} := \{b_1 + b_2a \bmod N \mid -B < b_1, b_2 < B\}$ and we say that $i + \mathcal{Q}$ or $(i + \bar{\mathcal{Q}})$ is a square centered on $i$. Observe that the elements of $i + \mathcal{Q}$ need to be placed at the square $ia + b + \bar{\mathcal{Q}}$ and we show that this can be done with an optimal number of block reads and writes. Then, the algorithm simply iterates over all the elements, and as long as there is an output position $j$ that needs to be filled, it reads a square from the input and writes it to the output such that it fills the position $j$. The non-trivial part of the analysis is to show that this runs in $O(N/B)$ I/Os.

**Lemma 3.1.** *The set of elements stored at index set $i + \mathcal{Q}$ can be read in $O(|\mathcal{Q}|/B)$ I/Os and be written to the indices defined by the set $ia + b + \bar{\mathcal{Q}}$ in $O(|\mathcal{Q}|/B)$ I/Os.*

*Proof.* We can view $\mathcal{Q}$ and symmetrically $\bar{\mathcal{Q}}$ as a union of $2B - 1$ intervals of length $2B - 1$. Their union can be found by viewing the array as a circle and applying the greedy arc cover minimization algorithm of (Lee & Lee, 1984). By merging overlapping intervals, we avoid reading (respectively writing) the same element more than once and achieve the claimed bounds. □

The main challenge behind showing the $O(N/B)$ I/O bound is that the squares moved by the algorithm can overlap, and thus some elements might be loaded more than once into memory. However, we prove that for every element this will occur at most 4 times, resulting in the desired I/O complexity. For this, we need the following lemma.

**Lemma 3.2.** *For any five indices $i_1, \ldots, i_5$ with the property that $\cap_{j=1}^{5}(i_j + \mathcal{Q}) \neq \emptyset$, there exists two indices $j$ and $j'$, $1 \leq j < j' \leq 5$ such that $i_j \in i_{j'} + \mathcal{Q}$ and $i_{j'} \in i_j + \mathcal{Q}$.*

*Proof.* Let $x \in \bigcap_{j=1}^{5}(i_j + \mathcal{Q})$. Then for each $j$ there exists $q_j \in \mathcal{Q}$ such that $x = i_j + q_j$, and hence $i_j = x - q_j$.

For any $j \neq j'$ we therefore have $i_j - i_{j'} = q_{j'} - q_j$. Since $q_j \in \mathcal{Q}$, we can write $q_j = a^{-1}b_j + c_j$ with $b_j, c_j \in [-B + 1, B - 1]$. Note that this representation might not be unique, but in that case, simply pick some $b_j$ and $c_j$ that satisfy the above. We also say $b_j$ is *positive* if $b_j \in [0, B-1]$ and negative otherwise (same for $c_j$).

Consider the five pairs $(b_j, c_j) \in [-B+1, B-1]^2$ for $1 \leq j \leq 5$. Observe that there exist two indices $1 \leq j < j' \leq 5$ such that both the pair $b_j$ and $b_{j'}$ and the pair $c_j$ and $c_{j'}$ have the same sign. This implies that $|b_j - b_{j'}| < B$ and $|c_j - c_{j'}| < B$.

It follows that

$$q_j - q_{j'} = a^{-1}(b_j - b_{j'}) + (c_j - c_{j'}) \in \mathcal{Q},$$

and hence $i_{j'} \in i_j + \mathcal{Q}$ and similarly, $i_j \in i_{j'} + \mathcal{Q}$. $\qquad \square$

**Theorem 3.1.** *Algorithm 1 creates 2-wise independent permutations in $O(N/B)$ I/Os.*

*Proof.* The algorithm implements the affine permutation $f_{a,b}(i) = ai + b \bmod N$ by first computing the multiplicative mapping $i \mapsto ai \bmod N$ within the main loop, and then applying the additive offset $b$ as a shift of the output array. Now as $f_{a,b}$ represents an affine transformation over a finite field, for $N$ prime, 2-wise independence follows from (Alon & Lovett, 2012).

The difficulty is in getting the I/O bound. The algorithm iteratively builds a union of squares $\bigcup_{j=1}^{\ell} (i_j + \mathcal{Q})$ by choosing a square center $i_\ell \notin \bigcup_{j=1}^{\ell-1} (i_j + \mathcal{Q})$ in iteration $\ell$. By Lemma 3.2, each element of the array can belong to at most 4 such squares.

Consider the total number of elements processed over all iterations, $|\bigcup_j (i_j + \mathcal{Q})| \leq \sum_j |i_j + \mathcal{Q}|$. Since each element appears in at most 4 such squares, we have $\sum_j |i_j + \mathcal{Q}| \leq 4N$. The I/O cost of processing these likewise becomes $\sum_j O(|i_j + \mathcal{Q}|/B) = O(N/B)$ by Lemma 3.1. $\qquad \square$

## 4. I/O efficient almost $k$-wise Permutations

In this section, we present our simple permutation algorithm that can generate almost $k$-wise permutations. The algorithm runs in two identical phases. In each phase, we load $B$ blocks (using $B$ I/Os). As each block is loaded, we uniformly shuffle its $B$ elements in internal memory. We then view the resulting $B^2$ elements as a $B \times B$ matrix, transpose the matrix, and write it to the output, shuffling newly created blocks again. This is efficient under the tall-cache assumption $M \geq B^2$, since the entire matrix fits in memory during transposition. We will use $n = N/B$ and $m = M/B$.

The intuition for why this produces an almost uniform shuffle is as follows. After the first phase, elements that started in different input blocks have been mixed essentially uniformly, while the main remaining dependence is between elements that started in the same block, since they were only shuffled locally. The transpose spreads each original block across many different blocks, so in the second phase, those within-block dependencies are largely broken as well.

When $N$ is much larger than $M$ and $B$, it is unlikely that two elements that start in the same block remain close across both phases (for example, by repeatedly landing in the same block again), so overall the permutation is close to uniform. We present the pseudocode in Algorithm 2 and a visualization in Figure 1.

---

**Algorithm 2** IO SHUFFLE($\mathcal{C}$)

1: Let $\mathcal{L} = (B_1, \ldots, B_n)$ be the list of $n = \lceil N/B \rceil$ blocks that store $\mathcal{C}$.
2: **for** $i = 1, 2$ **do**
3:     Randomly permute the blocks $\mathcal{L}$ and create an empty list $\mathcal{L}'$
4:     **while** $\mathcal{L}$ is non-empty **do**
5:         Remove the first $B$ blocks $B_1, \ldots, B_B$ from $\mathcal{L}$.
6:         Shuffle the elements of each $B_i$, $i = 1, \ldots, B$ uniformly at random.
7:         Form a $B \times B$ matrix $X = [B_1, \ldots, B_B]$ where each block forms a row and compute the transpose $Y = X^T$.
8:         For each row of $Y$, shuffle it uniformly at random and append to $\mathcal{L}'$
9:     **end while**
10:     Set $\mathcal{L} = \mathcal{L}'$ and clear $\mathcal{L}'$
11: **end for**
12: Shuffle the blocks of $\mathcal{L}$ uniformly at random
13: **return** Flatten($\mathcal{L}$)

---

**Lemma 4.1.** *Algorithm 2 performs $\mathcal{O}(N/B)$ IOs. In particular, it reads and writes each element twice.*

*Proof.* The proof is straightforward, see appendix A for details. $\qquad \square$

In the remainder of this section, we denote the permutation created by the above algorithm as $\pi$. Note that $\pi$ is a random variable. Let $\mu$ denote the distribution created by the algorithm on the set of all permutations.

Showing that the algorithm creates an almost $k$-wise permutation is more involved. Consider a distinct $k$-sequence $S = \langle x_1, \ldots, x_k \rangle$. We call the blocks that contain these elements *source blocks*. Recall that the algorithm does two iterations on the for loop, followed by a random shuffling of the blocks. Let $I = \langle i_1, \ldots, i_k \rangle$ be a distinct $k$-sequence of $[N]$ which describes the position of the elements $x_1, \cdots, x_k$ after the first iteration of the for loop. Note that these positions are random variables that depend on the random choices made by the algorithm in Lines 3,6,8. We also call the set of $B$ blocks read at Line 5 of the algorithm a *gate*. Thus, each gate reads $B$ input blocks and writes $B$ output blocks. A gate is responsible for a target position if the gate writes an element to the target position.

**Lemma 4.2.** *Let $E_1$ be the event that two positions in $I$ belong to the same block. We have $\Pr[E_1] \leq \frac{k^2 B}{n}$.*

*Proof.* See Appendix A. $\qquad\square$

**Theorem 4.1.** *For any distinct $k$-sequence $S$ of the elements and any distinct $k$-sequence $I$ of $[N]$ we have*

$$\Pr[\pi(S) = I] \geq \frac{\left(1 - \frac{2k^2 B}{n}\right)}{N(N - B)\cdots(N - (k-1)B)}$$

*Proof.* See Appendix A. $\qquad\square$

**Theorem 4.2.** *For any distinct $k$-sequence $S$ of the elements and any distinct $k$-sequence $I$ of $[N]$ we have $\mu_S(I) \geq \left(1 - \frac{2k^2 B}{n}\right) \mathfrak{U}_S(I)$.*

*Proof.* See Appendix A for details. $\qquad\square$

The above theorem, in turn, implies that our distribution is almost $k$-wise independent.

**Theorem 4.3.** *Algorithm 2 generates almost $k$-wise independent permutatations with total error of $\frac{4k^2 B}{n}$.*

*Proof.* Let $\mathcal{I}$ be the set of all the distinct $k$ sequences of $[N]$ and let $S$ be a fixed distinct $k$-sequence of the input elements. Let $\varepsilon = \frac{2k^2 B}{n}$. We partition $\mathcal{I}$ into two sets, $\mathcal{I}_1$ and $\mathcal{I}_2$, such that for any sequence $I \in \mathcal{I}_1$ we have $\mu_S(I) \geq \mathfrak{U}_S(I)$ and for any sequence $I \in \mathcal{I}_2$ we have $\mu_S(I) < \mathfrak{U}_S(I)$. Let $m_i = \sum_{I \in \mathcal{I}_i} \mu_S(I)$ and $u_i = \sum_{I \in I_i} \mathfrak{U}_S(I)$ for $i = 1, 2$. Observe that we have $m_1 + m_2 = u_1 + u_2 = 1$ as $\mu$ and $\mathfrak{U}$ are probability distributions. Additionally, by Theorem 4.2, we get the inequality that $u_2(1 - \varepsilon) \leq m_2 \leq u_2$. Observe that by our notation, the distance between the distributions $\mu_S$ and $\mathfrak{U}_S$ becomes $u_2 - m_2 + m_1 - u_1 = 2(u_2 - m_2)$ which by the above inequality is bounded by $2u_2 \leq 2\varepsilon$. $\quad\square$

## 5. The Necessity of Tall-Cache

In this section, we will prove that even creating almost two-wise independent permutations requires the tall cache assumption ($M = \Omega(B^2)$). We note that the tall-cache assumption is very common in real-life systems: as an example, consider the cache-memory hierarchy. The size of the cache line for most processors is 64 or 128 bytes (e.g., Intel i5 uses 64 bytes, an SM core in Nvidia GH200 uses 128 bytes): the tall cache assumption is satisfied when the total cache size is at least 16KB, and typically L1/L2 cache sizes by now far exceed these values (e.g., 12 MB for i5, 128KB for GH200). In the memory-disk hierarchy, typical block sizes are 4-16KB and the tall cache assumption is satisfied with only 256MB of memory.

From a theoretical perspective, previous works have investigated the necessity of the tall-cache for optimal cache-oblivious algorithms for sorting (Brodal & Fagerberg, 2003) and some permutation families (Silvestri, 2008).

Our model of computation is the *indivisibility model* (Aggarwal & Vitter, 1988) where the input $\mathcal{C} = a_1, \cdots, a_N$ is a sequence of atomic elements and the goal is to generate a permutation $\pi$ of $\mathcal{C}$ where the element $a_i$ is placed at position $\pi(i)$. The input elements sit on a disk of conceptually unbounded capacity, and the atomicity of the elements means that the algorithm can read a block of $B$ atomic elements from the disk into the main memory of size $M$, or conversely, select $B$ elements from those present in the internal memory and write them as a block to the disk. Also, since the algorithm is creating a permutation of the elements, it thus suffices to consider algorithms that only move the elements and do not create new ones, as any extra copies that end up not participating in $\pi$ can be deleted (or not created during the algorithm).

**Lemma 5.1.** *Let $\mathcal{C} = a_1, \cdots, a_N$ be an input sequence of $N$ atomic elements, stored in $n = N/B$ input blocks. Assume that we have obtained a permutation $\pi$ of $\mathcal{C}$ using $\frac{\tau N}{B}$ I/Os for a parameter $\tau > 0$, and $\pi$ is stored in $n$ output blocks. Then, there exists a set $B_{\mathcal{C}}$ containing $\Omega(n)$ input blocks, such that for every input block $b \in B_{\mathcal{C}}$, there exists an output block $b'$ such that $b$ and $b'$ have at least $\frac{B}{m^{\mathcal{O}(\tau)}}$ elements in common where $m = \frac{M}{B}$.*

*Proof.* See appendix B $\qquad\square$

We now state the main result of this section.

**Theorem 5.1.** *Assuming $N \geq B^2$, there exists a fixed constant $\delta' > 0$ such that for any constant $\delta < \delta'$, any randomized algorithm $\mathcal{A}$ that generates an almost $2$-wise independent permutation of a set $\mathcal{C}$ of $N$ elements with total error $\varepsilon$ either uses $\frac{\delta N \log_m B}{B}$ I/Os on average or has total error $\epsilon = \Omega(B^{-\mathcal{O}(\delta)})$.*

*Proof.* See appendix B $\qquad\square$

## 6. SGD convergence with $2$-wise permutations

For the purposes of this paper, we limit our analysis to the strongly convex regime, and prove the following theorem[2]. For simplicity, we will assume that all functions $f_i$ and hence $F$ are defined on some domain $\mathcal{W} \subset \mathbb{R}^d$. As in previous works, we make the following assumptions: strong

---

[2]We also remark that in the low epoch regime, the bounds in (Nagaraj et al., 2019) are better than those in (Haochen & Sra, 2019). However, as far as we can see, the proof in (Nagaraj et al., 2019) does not seem to hold for 2-WISE-RANDOMSHUFFLE.

*Table 2.* Block-level clustering statistics for $N = 16\,000\,057$ and block size $B = 400$ over $40\,000$ repetitions. We report the median number of preserved pairs with quantiles (Q1–Q3), the mean is shown to illustrate tail behavior. The expectation for fully random shuffle is $\mathbb{E}[X] = \frac{16\,000\,057}{400}\binom{400}{2} \cdot \frac{400-1}{16\,000\,057-1} \approx 79\,600.5$ supporting the claim that 2 rounds of IO Shuffle is very close to uniform.

| Algorithm | Median | Q1–Q3 | Mean |
|---|---|---|---|
| Gen-2-Wise-Ind-Perm | 0 | 0–0 | 74 305 |
| IO Shuffle (1 round) | 60 591 | 60 591–60 591 | 60 589 |
| IO Shuffle (2 rounds) | 79 598 | 79 335–79 867 | 79 603 |
| CorgiPile (1 round) | 8 017 871 | 8 015 719–8 019 986 | 8 017 852 |
| Fisher–Yates | 79 600 | 79 412–79 790 | 79 601 |

convexity, diameter bound, Lipschitz Continuity, Smoothness, Hessian Lipschitz. We refer to Appendix C for the definitions.

**Theorem 6.1.** *Assume that to solve the minimization problem for $F(x)$, we use algorithm* 2-WISE-RANDOMSHUFFLE *instead of* RANDOMSHUFFLE. *Then, under the same respective assumptions, all the results in (Gürbüzbalaban et al., 2021) and (Haochen & Sra, 2019) (see Table 1) hold for* 2-WISE-RANDOMSHUFFLE.

In light of Algorithms 1 and 2, the theorem above shows that up to a small constant, *only a linear number of I/Os are enough to guarantee that* 2-WISE-RANDOMSHUFFLE *beats SGD in expectation*. The proof of Theorem 6.1 can be found in the Appendix C. It requires a careful analysis of the proofs in the papers (Gürbüzbalaban et al., 2021; Haochen & Sra, 2019). We replace the fully random permutation $\sigma_e$ in epoch $e$ with a permutation drawn from a family of 2-wise independent permutations, and derive the same results under the weaker notion of 2-wise independence.

$k$**-wise independence and** **K-WISE-RANDOMSHUFFLE.** While the above result shows that one can replicate the theoretical guarantees for RANDOMSHUFFLE with only 2-WISE-RANDOMSHUFFLE, there could be scenarios where 2-WISE-RANDOMSHUFFLE is not enough, and one may need K-WISE-RANDOMSHUFFLE, especially to obtain high probability bounds. We state an example with a high probability first-order guarantee where this is likely the case.

In (Yu & Li, 2023), the authors provide a high probability guarantee under the additional assumption that for all $i \in [N]$, each function $f_i$ is lower bounded by $\bar{f}_i$. Their result states that when $T = O(\max\{\sqrt{N}\epsilon^{-3}, N\epsilon^{-2}\} \cdot \log(\sqrt{N}\epsilon^{-3}/\delta))$, $\frac{1}{\ell}\sum_{i=1}^{\ell}||\nabla f(x_i^0)||^2 \le \epsilon^2$ with probability at least $1 - \delta$. In the Appendix C.3, we sketch how the results in (Yu & Li, 2023) can be adapted to K-WISE-RANDOMSHUFFLE instead of random shuffle.

## 7. Experiments

To test how close our algorithms are to 2-wise independence, we implement them in Python[3] and make the simple proof-of-concept experiment of counting how many pairs of indices that start within the same block also end within the same block. This is a good proxy for 2-wise independence because block structure is exactly the granularity at which I/O efficiency and memory locality are determined: correlations that cause pairs of indices to remain in the same block indicate insufficient mixing at the level relevant to external-memory access, even if element-level positions appear random. Under a uniformly random permutation, the expected number of unordered pairs of indices that start in the same input block and end in the same output block is $\mathbb{E}[X] = \frac{N}{B}\binom{B}{2} \cdot \frac{B-1}{N-1}$.

Table 2 shows clear differences between shuffling strategies. The uniformly random permutation algorithm FISHER-YATES matches the expected baseline. In contrast, CORGIPILE retains a large number of pairs that start and end in the same block, even after shuffling. As expected, our GEN-2-WISE-IND-PERM and IO SHUFFLE algorithms efficiently eliminate such initial clustering.

## 8. Conclusion and Future Work

In this paper, we have studied to what extent random permutations can be generated with a limited amount of I/Os. Our results have applications in improving the performance of Stochastic Gradient Descent for strongly convex loss functions: we have indeed shown that we can achieve an optimal I/O cost (i.e., $\Theta(N/B)$ I/Os) *and* also $O(1/T^2)$ convergence as in the RANDOMSHUFFLE algorithm. Our results are strictly theoretical, and an interesting open question is to experimentally evaluate our findings. Another direction is to derive 2-wise or (almost) $k$-wise algorithms that are cache-oblivious (Frigo et al., 2012), which are independent of the I/O model parameters, $B$ and $M$. Finally, we leave future work to understand how generating *almost* $k$-uniform permutations changes the convergence of SGD.

---

[3]The code can be found at this github repository.

## Acknowledgments

This work was supported in part by the DFF (Danmarks Frie Forskningsfond) of the Danish Council for Independent Research under grant ID 10.46540/3103-00334B, and by the U.S. National Science Foundation (NSF) grants CCF-2318633 and CCF-2503086. This material is based upon work performed while attending AlgoPARC Workshop on Parallel Algorithms and Data Structures at the University of Hawaii at Manoa, in part supported by the National Science Foundation under Grant No. 2452276.

## Impact Statement

This paper presents work whose goal is to advance the field of machine learning. There are many potential societal consequences of our work, none of which we feel must be specifically highlighted here.

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

## A. Omitted proofs from section 4

*Proof of Lemma 4.1.* First, we analyze the number of I/Os. Creating a random permutation of the blocks can be done with $N/B$ I/Os using, for instance, the Fisher-Yates algorithm. However, this step is not strictly necessary. In particular, we can simply create a random permutation of *indices* of the blocks, i.e., the numbers 1 to $N/B$, which can be done with $\min\{N/B, O(\frac{N}{B^2}\log_B(N/B^2))\}$ I/Os. Thus, to perform Step 5, we simply read the next blocks of indices that contain the indices of the next blocks and then we read the blocks themselves. Steps 7 and 8 can be done in memory. Similar to Step 5, Step 12 can be done implicitly by simply writing each block where it is supposed to go during the second iteration of Step 8. Thus, in total, each element is read and written twice. $\square$

*Proof of Lemma 4.2.* To derive this result, it suffices to bound the probability that each gate places the elements it is responsible for in distinct blocks. Let $E_1'$ be the event that at least one gate receives at least two source blocks. We claim $\Pr[E_1] \leq \Pr[E_1']$ since the elements that are in the same block will be placed in different output blocks because, in Line 7, we are performing a matrix transpose. Observe that a random permutation of $n$ elements can be created by starting with an empty array of size $n$ and then iterating over the elements and placing each element in a random empty position in the array. To bound the probability of $E_1'$, we consider this process by iterating over the source blocks. $E_1'$ is the event that, during this process, we place a source block at an empty position that belongs to a gate that already has a source block assigned to it. Since the number of source blocks is at most $k$, this probability is at most $\frac{kB}{n}$. Using a union bound, we get that $\Pr[E_1'] \leq \frac{k^2B}{n}$. $\square$

**Theorem 4.1.** *For any distinct $k$-sequence $S$ of the elements and any distinct $k$-sequence $I$ of $[N]$ we have*

$$\Pr[\pi(S) = I] \geq \frac{\left(1 - \frac{2k^2B}{n}\right)}{N(N-B)\cdots(N-(k-1)B)}$$

*Proof.* Assume the event $E_1$ does not hold. By lemma 4.2, this happens with probability $1 - \frac{k^2B}{n}$. We now consider the second iteration of the for loop. Consider a particular outcome of the first iteration of the for loop and, as event $E_1$ does not happen, the elements of $S$ are spread over distinct blocks at the beginning of the second iteration. We call the blocks that contain the elements of $S$ the start of the second iteration of the for loop the *secondary source blocks*.

To reduce the number of notations, we will reuse some of the previously defined notations. Let $I = \langle i_1, \ldots, i_k \rangle$ be a distinct $k$-sequence of $[N]$, and we will lower bound the probability that the elements of $S$ are placed in positions given by $I$. We call these the *target positions*, and the block that contains any target position a *target block*. Observe that the last step of the algorithm involves another random permutation of the blocks. This is a random process that is independent of all the other steps of the algorithm; let $\mathfrak{P}$ denote the random variable that describes the random permutation that is used here. Obviously, for any fixed permutation $P$, we have $\Pr[\mathfrak{P} = P] = \frac{1}{n!}$, as there are $n = N/B$ blocks. Also, observe that any fixed permutation $P$ maps the target positions to the gates (in the second iteration of the for loop) and that some of the gates will be responsible for generating these target positions. We say $P$ is *good* if each gate is responsible only for one target block, otherwise it is *bad*. Observe that a bad permutation has two target blocks assigned to it, and thus we can bound the probability of obtaining a bad permutation by Lemma 4.2 as well. Let $E_2$ denote the event that $P$ is a bad permutation.

Assume that $\mathfrak{P} = P$ for some good permutation $P$. This, in turn, fixes the target positions, and thus each gate is responsible for some of the target positions. We ignore the gates that are not responsible for any target positions. Thus, assume we have $g$ gates that are responsible for at least one target position; with a slight abuse of notation, number these gates from 1 to $g$ and assume they are responsible for $k_1, \ldots k_g$ target positions. Also, since $P$ is good, each gate is responsible only for one target block. We now calculate the probability of the event claimed in the lemma. Since we are assuming $\neg E_1$, all the source positions are in distinct blocks. Thus, the $i$-th gate must receive exactly the $k_i$ blocks that have the correct element that must be placed in the $k_i$ target locations produced by the $i$-th gate. This is determined by the permutation produced in Line 3 during the 2nd iteration of the for loop. Any permutation that produces such an ordering is called *useful*.

To bound this probability, observe that the total number of permutations of the blocks is $n!$. Thus, it remains to count the number of useful permutations. To do that, fix a permutation of the $n - k$ non-source blocks ($(n-k)!$ choices), and then for the $i$-th gate, select the positions of $k_i$ source blocks that belong to the $i$-th gate, in an arbitrary permutation. For the $i$-gate, this gives $\binom{B}{k_i}k_i!$ choices. Thus, the probability of getting a good permutation is

$$\frac{(n-k)!\prod_{i=1}^{g}\binom{B}{k_i}k_i!}{n!} \tag{1}$$

These blocks form the rows of the matrix $X$ formed in Line 6. Each row of $X$ contains either one or zero source elements. If a row contains a source element, it must be per-

muted to the correct index, which happens with probability $1/B$ and thus with $1/B^{k_i}$ over all the $k_i$ source elements. Finally, after producing the matrix $Y$, in Line 8 we must place the source elements in the correct target locations. As $P$ is good, each gate produces only a single target block, and thus this probability is $\frac{1}{B(B-1)\ldots(B-k_i+1)}$ simply because we are producing a uniform random permutation of the elements in the block, and thus we can consider that the $t$-th source element goes to the correct location with probability $\frac{1}{B-t+1}$. Putting all this together, we get the following:

$$\frac{(n-k)! \prod_{i=1}^{g} \binom{B}{k_i} k_i!}{n!} \cdot \frac{1}{\prod_{i=1}^{g} B^{k_i}}$$
$$\cdot \prod_{i=1}^{g} \frac{1}{B\ldots(B-k_i+1)}$$

We can simplify the expression to the following:

$$\frac{\prod_{i=1}^{g} B(B-1)\ldots(B-k_i+1)}{n(n-1)\ldots(n-k+1)} \cdot$$
$$\frac{1}{B^k} \cdot \prod_{i=1}^{g} \frac{1}{B(B-1)\ldots(B-k_i+1)}$$

which, by the union bound, yields the probability claimed in the lemma. □

**Theorem 4.2.** *For any distinct $k$-sequence $S$ of the elements and any distinct $k$-sequence $I$ of $[N]$ we have $\mu_S(I) \geq \left(1 - \frac{2k^2 B}{n}\right) \mathfrak{U}_S(I)$.*

*Proof.* Observe that $\mathfrak{U}_S(I)$ is the uniform distribution and thus we have

$$\mathfrak{U}_S(I) = \frac{1}{N \cdot (N-1) \cdot \ldots \cdot (N-k+1)}.$$

We consider the ratio $\frac{\mu_S(I)}{\mathfrak{U}_S(I)}$ and use Theorem 4.1 to obtain

$$\frac{\mu_S(I)}{\mathfrak{U}_S(I)} \geq \left(1 - \frac{2k^2 B}{n}\right) \prod_{i=0}^{k-1} \frac{N-i}{N-iB}$$
$$\geq \left(1 - \frac{2k^2 B}{n}\right)$$

where the last step follows from the fact that $N - i \geq N - iB$. □

## B. Omitted proofs from section 5

*Proof of Lemma 5.1.* The proof uses two classical ideas, one is a reduction by (Jia-Wei & Kung, 1981), and the other is the potential function argument of Aggarwal and Vitter (Aggarwal & Vitter, 1988) which in turn was based on a potential function argument by Floyd (Floyd, 1972).

By (Jia-Wei & Kung, 1981), we can assume that $\pi$ is obtained by an algorithm that works in phases, where in each phase the algorithm reads $m$ blocks and then writes a permutation of their elements back to the disk (using $m$ block writes) and then flushes the memory (i.e., no elements are kept in the memory). Any arbitrary algorithm can be transformed into such an algorithm by only a constant blow-up in the I/O complexity.

Now, we use the potential function argument. For an input element $a_i$, consider the output block $b$ of $\pi$ that contains $a_i$. Next, consider the state of the disk at the end of the $j$-th phase of the algorithm; as the memory is empty, the elements on the disk are now in a permutation $\mathcal{C}_j$, and let $b'$ be the block that contains $a_i$. The potential of $a_i$ at the end of step $j$ is defined as $\phi_j(a_i) = \log(|b \cap b'|)$, i.e., the logarithm of the number of elements that are in common between the current block of $a_i$ and its output block (including $a_i$ itself). The overall potential at the end of step $j$ is defined as $\phi_j = \sum_{i=1}^{N} \phi_j(a_i)$. The main point is that the increase in the potential from phase $j$ to phase $j+1$ is bounded:

$$\phi_{j+1} \leq \phi_j + M \log m. \tag{2}$$

To see this, consider phase $j$ during which the algorithm first reads a set $S$ of $M$ elements in $m$ blocks $b_1, \cdots, b_m$ and then writes $m$ blocks, $b'_1, \ldots, b'_m$, which contain a permutation of $S$ to the disk. Consider one input block $b$ and assume $b$ has $\ell_k(b)$ elements in $b_k$, and $\ell'_k(b)$ elements in $b'_k$, $1 \leq k \leq m$. Let $\ell(b) = \sum_{k=1}^{m} \ell_k(b) = \sum_{k=1}^{m} \ell'_k(b)$. Also observe that $\sum_b \ell(b) = \sum_b \ell'(b) = M$ since at every phase, $M$ elements are in the memory. The total contribution of elements in $S$ to the potential function $\phi_j$ at step $j$ is

$$E_j = \sum_b \sum_{k=1; \ell_k(b)>0}^{m} \ell_k(b) \log(\ell_k(b)) \tag{3}$$

and at step $j+1$ is

$$E_{j+1} = \sum_b \sum_{k=1; \ell'_k(b)>0}^{m} \ell'_k(b) \log(\ell'_k(b)). \tag{4}$$

Due to the convexity of the $x \log x$ function, and under the constraint that $\ell(b)$ is the sum of all $\ell_k(b)$ and $\ell'_k(b)$ values, Eq. (3) is minimized when all $\ell_k$ are equal but Eq. (4) is maximized when one $\ell'_k$ contains all the mass, meaning,

$$E_j \geq \sum_b \sum_{k=1; \ell(b)>0}^{m} \frac{\ell(b)}{m} \log\left(\frac{\ell(b)}{m}\right)$$

and

$$E_{j+1} \leq \sum_{b; \ell(b)>0} \ell(b) \log(\ell(b)).$$

These two facts imply that

$$E_{j+1} - E_j \leq \sum_{b; \ell(b) > 0} \ell(b) \log(m) \leq M \log m.$$

Since the sum of $\ell(b)$ over all input blocks is at most $M$ and thus Eq. 2 holds.

Now we have all the ingredients necessary to prove the lemma. Let $f$ be the number of phases. Observe that since we have $\frac{\tau N}{B}$ I/Os, we get that $f = O\left(\frac{\tau N}{M}\right)$. Let $\phi_0$ be the starting potential and $\phi_f$ be the final potential (at the end of the last phase). Thus,

$$\phi_f - \phi_0 \leq f \cdot M \log m \leq O\left(\tau N \log m\right) \leq c\tau N \log m$$

for some constant $c > 0$. As a result, there are at least $N/2$ elements $a_i$ whose potential increases by at most $2c\tau \log m$, meaning, $\phi_f(a_i) - \phi_0(a_i) \leq 2c\tau \log m$. However, the way the potential function is defined, we have $\phi_f(a_i) = \log B$. This means that $\phi_0(a_i) \geq \log B - 2c\tau \log m$ which means the input block containing $a_i$ has at least $\frac{B}{m^{2c\tau}}$ elements in common with the output block containing $a_i$. As there are $N/2$ such input elements $a_i$, it follows that there are $\Omega(n)$ blocks as claimed in the lemma. $\square$

**Theorem 5.1.** *Assuming $N \geq B^2$, there exists a fixed constant $\delta' > 0$ such that for any constant $\delta < \delta'$, any randomized algorithm $\mathcal{A}$ that generates an almost 2-wise independent permutation of a set $\mathcal{C}$ of $N$ elements with total error $\varepsilon$ either uses $\frac{\delta N \log_m B}{B}$ I/Os on average or has total error $\epsilon = \Omega(B^{-\mathcal{O}(\delta)})$.*

*Proof.* If the algorithm uses at least $K = \delta \frac{N \log_m B}{B}$ I/Os then there is nothing to prove. Thus, assume that the algorithm does at most $K$ I/Os in expectation, and now we do a proof by contradiction using a double-counting argument. Let $\mu$ be the random distribution of the permutations of $\mathcal{C}$ created by $\mathcal{A}$. We first consider the input $\mathcal{C}$ and select a random block $\mathbf{b}$ uniformly at random. Then, we select two distinct random elements, $\mathbf{a}$ and $\mathbf{a}'$, of $\mathbf{b}$ uniformly at random. Next, we pick a permutation $\pi$ according to $\mu$. Let $B$ and $B'$ be the blocks of $\pi$ that contain the elements $\mathbf{a}$ and $\mathbf{a}'$. Define

$$X = \Pr[B = B'],$$

i.e., the event that $\mathbf{a}$ and $\mathbf{a}'$ are in the same block in $\pi$.

First, observe that since $\mu$ is an almost 2-wise independent distribution with total error $\varepsilon$, for any two elements $a$ and $a'$, the probability that they end up in the same block is at most

$$X \leq \varepsilon + \frac{1}{n} \tag{5}$$

since in the uniform distribution, $a$ and $a'$ will be in the same block with probability at most $\frac{1}{n}$ (to see this, assume $a$ is placed first at a random position and consider

the placement of $a'$; $a'$ will be placed at the same block as $a$ with probability $\frac{B-1}{N} < \frac{1}{n}$). By the Markov inequality, with probability at least $\frac{1}{2}$, $\pi$ is generated using at most $2K$ I/Os. Pick a permutation $\pi$ that is created using at most $2K$ I/Os. By Lemma 5.1, and by choice of $\mathbf{b}$, with $\Omega(1)$ probability, the random variable $\mathbf{b}$ will be a block $b$ such that there exists a block $b'$ in $P$ such that $b$ and $b'$ have at least $r = \frac{B}{m^{\mathcal{O}(\delta \log_m B)}} = \frac{B}{B^{\mathcal{O}(\delta)}} = B^{1-\mathcal{O}(\delta)}$ elements in common. Observe that with probability at least $\frac{r(r-1)}{2B(B-1)} = \Omega(B^{-\mathcal{O}(\delta)})$, both $\mathbf{a}$ and $\mathbf{a}'$ are chosen among these common $r$ elements, then for them it holds that they are placed in the same block after the permutation.

$$X > \Omega(B^{-\mathcal{O}(\delta)}). \tag{6}$$

Let $c$ be the constant hidden in the $\Omega$ notation above. By picking $\delta$ small enough, we can ensure that $\frac{1}{n} < \frac{1}{2B^{-c\delta}}$ which ensures that

$$\varepsilon > \frac{1}{2B^{-c\delta}}$$

proving our theorem. $\square$

## C. Omitted proofs from section 6

We consider the known results in chronological order, and prove the analogous versions for 2-WISE-RANDOMSHUFFLE and k-WISE-RANDOMSHUFFLE. We will have to deviate from the respective notation in the previous papers, because $k$ in this work stands for the level of independence.

For simplicity, we will assume that all function $f_i$ and hence $F$ are define on some domain $\mathcal{W} \subset \mathbb{R}^d$. As in the previous papers, we make the following assumptions:

**Assumption 1.** (Strong Convexity) There exists $\mu > 0$ such that $F(y) \geq F(x) + \langle \nabla F(x), y - x \rangle + \frac{\mu}{2} ||y - x||^2$ for all $x, y \in \mathcal{W}$.

**Assumption 2.** (Diameter Bound) All iterates $x$ satisfy $||x - x^*|| \leq D$ for some $D > 1$.

**Assumption 3.** (Lipschitz Continuity) There exists $G > 0$ such that $||\nabla f_i(x)|| \leq G$ for all $i \in [n]$ and all $x \in \mathcal{W}$.

**Assumption 4.** (Smoothness, or Lipschitz Gradient) There exists $L > 0$ such that $||\nabla f_i(x) - \nabla f_i(y)|| \leq L ||x - y||$ for all $i \in [n]$ and all $x, y \in \mathcal{W}$.

**Assumption 5.** (Hessian Lipschitz) Let $H_i(x)$ denote the Hessian of the function $f_i$ at $x$. For each $i \in n$ and for all $x, y \in \mathcal{W}$ there exists a constant $L_H^i$ such that $||H_i(x) - H_i(y)|| \leq L_H^i ||x - y||$.

## C.1. Results by (Gürbüzbalaban et al., 2021)

The results in (Gürbüzbalaban et al., 2021) are first derived for the quadratic case, and then extended to the strongly convex case. We show that any Theorems/Lemmata in (Gürbüzbalaban et al., 2021) that depend on $\sigma_e$ being uniformly randomly chosen, also apply when $\sigma_e$ is only chosen from a 2-wise independent distribution.

To this end, let us first examine the proof in the quadratic case. So for $1 \leq i \leq n$, the functions $f_i : \mathbb{R}^d \to \mathbb{R}$ are of the form

$$f_i(x) = \frac{1}{2} x^T P_i x_i + q_i^T x + r_i,$$

where $P_i$ is a symmetric $d \times d$ matrix, $q_i$ is a vector in $\mathbb{R}^d$ and $r_i$ is a scalar. Note that $(.)^T$ denotes the transpose operator and should not be confused with the total number of iterations $T$.

Of crucial importance in the proof in (Gürbüzbalaban et al., 2021) is the following quantity (Equation 20):

$$\mu(\sigma) = - \sum_{1 \leq i < j \leq n} P_{\sigma(j)} \nabla f_{\sigma(i)}(x^*), \qquad (7)$$

where $x^*$ is the unique minimum of $F$ and $\sigma$ is an arbitrary permutation. Note that this $\mu$ is different from the strong-convexity parameter $\mu$ in Assumption 1: the strong-convexity parameter will not appear in the proof for the quadratic case. Here is a summary of the results in Sections 4.1 and 4.2 in (Gürbüzbalaban et al., 2021):

1. Theorem 2 in Section 4.1 derives a bound on the *expected* distance between the outer iterates $x_0^e$ and $x^*$. Instead of a worst case bound on $\mu(\sigma)$, they show that when $\sigma$ is randomly sampled, replacing $\mu(\sigma)$ by $\bar{\mu} := \mathbb{E}(\mu(\sigma))$ gives a factor $n$ improvement. The latter expectation is calculated in Lemma B.3 in the appendix in (Gürbüzbalaban et al., 2021), and is the only place where the random nature of $\sigma$ appears.

2. Theorem 3 in Section 4.2 derives a high-probability bound on the distance between the $q$-suffix iterates (these are averages of $x_0^e$ over the last $q\ell$ many epochs $e$, for some constant $0 < q < 1$) and $x^*$. At the heart of this (and the only place where the randomness of $\sigma$ appears) is Lemma B.4 in the appendix. Lemma B.4 shows that

$$\lim_{\ell \to \infty} \sum_{i=1}^{\ell} \frac{\mu(\sigma_{e_i})}{\ell} = \bar{\mu} \text{ almost surely,}$$

where $\mu(\sigma_{e_i})$ for $1 \leq i \leq \ell$ is an i.i.d. sequence of random variables.

We have the following claim, that shows that the quadratic case results in (Gürbüzbalaban et al., 2021) hold for 2-WISE-RANDOMSHUFFLE.

**Claim 1.** Lemma B.3 and B.4 in (Gürbüzbalaban et al., 2021) hold when $\sigma$ is drawn from a 2-wise independent distribution.

*Proof.* We first consider Lemma B.3 that computes $\bar{\mu} = \mathbb{E}(\mu(\sigma))$ and shows that this equals $(1/2) \sum_{i=1}^{n} P_i \nabla f_i(x^*)$. Consider the $(i, j)$-th term in $\mu(\sigma)$ (Equation 7). For Lemma B.3, the proof calculates $\mathbb{E}_\sigma(P_{\sigma(j)} \nabla_{\sigma(i)}(x^*)$, and since only two indices appear here, the calculation only requires that for $j \neq i$, the distribution of $(\sigma(j), \sigma(i))$ is uniformly random over all ordered pairs in $[n] \times [n]$. This, however, is the definition of 2-wise independence, and therefore 2-wise independence suffices for this lemma.

The proof of Lemma B.4 relies on the sequence $\mu(\sigma_{e_i})$ for $1 \leq i \leq \ell$ being independent and identically distributed, and then using the strong law of large numbers to show almost sure convergence. In our case, this sequence is clearly identically distributed, since in every epoch we draw $\sigma_{e_i}$ from the same 2-wise distribution over the space of permutations. However, we no longer have independence. To this end, we use the result by (Etemadi, 1981) that shows that the strong law of large numbers also applies when the random variables are only pairwise independent. This is exactly the setting for 2-WISE-RANDOMSHUFFLE, and so Lemma B.4 works here too. □

For the *non-quadratic, arbitrary strongly convex case*, all proofs follow similar arguments to the quadratic case, except now instead of $\mu(\sigma)$ (Equation 7) one has to consider the quantity:

$$\upsilon(\sigma) = - \sum_{i=1}^{n} \nabla^2 f_{\sigma(i)}(x^*) \sum_{j=1}^{i} \nabla f_{\sigma(k)}(x^*). \qquad (8)$$

Note that in the quadratic case $\nabla^2 f_{\sigma(i)}(x^*)$ was replaced simply with $P_i$. The authors then compute $\bar{\upsilon} := \mathbb{E}_\sigma[\upsilon(\sigma)]$, when $\sigma$ is a uniformly randomly chosen permutation. Again, because only two indices appear in the expectation computation, pairwise independence suffices, and the same result holds for 2-WISE-RANDOMSHUFFLE.

Next, the authors consider the i.i.d. sequence of random variables $\upsilon(\sigma_{e_i})$ for epochs $1 \leq i \leq \ell$, and apply the strong law of large numbers. Because of the result by (Etemadi, 1981), we know the same result applies under pairwise independence, and hence all results in the strongly convex case apply to 2-WISE-RANDOMSHUFFLE too.

## C.2. Results by (Haochen & Sra, 2019)

Similar to (Gürbüzbalaban et al., 2021), (Haochen & Sra, 2019) also first consider the quadratic case and then the strongly convex case. We skip the quadratic case and show how to adapt their proof for the strongly convex case to 2-WISE-RANDOMSHUFFLE instead of RANDOMSHUFFLE.

The randomness of the permutation $\sigma_e$ across epochs $e$ appears in proof via an error term $R^e$, defined as follows.

$$R^e = \sum_{i=1}^{n} \nabla f_{\sigma_e(i)}(x_{i-1}^e) - \sum_{i=1}^{n} \nabla f_{\sigma_e(i)}(x_0^e).$$

What is required in the proof is to bound $\mathbb{E}[||R^e||^2]$, and more crucially, bounding $\langle x_0^e - x^*, \mathbb{E}[R^e] \rangle$.

This error term can be decomposed into three quantities: $R^e = A^e + B^e + C^e$ as below, where $\gamma = 4\frac{\log(n\ell)}{\mu n\ell}$ is the step size, and $\mu$ is the strong-convexity parameter.

$$A^e = -\gamma \sum_{i=1}^{n} \left[ H_{\sigma_e(i)}(x^*) \sum_{j=1}^{i-1} \nabla f_{\sigma_e(j)}(x_0^e) \right],$$

$B^e =$

$$-\gamma \sum_{i=1}^{n} \left[ H_{\sigma_e(i)}(x^*) \sum_{j=1}^{i-1} \left( \nabla f_{\sigma_e(j)}(x_{j-1}^e) - \nabla f_{\sigma_e(j)}(x_0^e) \right) \right],$$

and

$$C^e = \sum_{i=1}^{n} \left[ \int_{x_0^e}^{x_{i-1}^e} (H_{\sigma_e(i)}(x) - H_{\sigma_e(i)}(x^*)) dx \right].$$

Now, when it comes to computing $\langle x_0^e - x^*, \mathbb{E}[R^e] \rangle$, (Haochen & Sra, 2019) only really compute the expectation of $\langle x_0^e - x^*, \mathbb{E}[A^e] \rangle$; for $B^e$ and $C^e$ they bound the norms $||B^e||$ and $||C^e||$, and use the AM-GM inequality to move from bounding $\langle x_0^e - x^*, \mathbb{E}[B^e] \rangle$ and $\langle x_0^e - x^*, \mathbb{E}[C^e] \rangle$ to terms involving $||B^e||$ and $||C^e||$ and using the derived upper bounds.

While bounding $||B^e||$ and $||C^e||$, the only place where randomness of $\sigma$ is used is the claim that $\mathbb{E}_{i,j} H_{\sigma(i)} H_{\sigma_j} = H^2$ (here $H$ is the Hessian of $F$). Because $(\sigma(i), \sigma(j))$ is uniformly distributed across $[n] \times [n]$, this holds for 2-WISE-RANDOMSHUFFLE too.

The second difference in the proof for 2-WISE-RANDOMSHUFFLE versus RANDOMSHUFFLE is the computation of $\mathbb{E}(A^e)$. Rearranging the terms in the definition, one sees that $A^e$ is a sum of $n(n-1)/2$ many terms, each with the same expectation, so we have

$$\mathbb{E}(A^e) = -\frac{n(n-1)\gamma}{2} \mathbb{E}_{i \neq j}[H_{\sigma_e(i)} \nabla f_{\sigma_e(j)}(x_0^t)].$$

Due to the same argument as in Equation 7, namely that for the above computation one only requires the distribution of $(\sigma(i), \sigma(j))$ to be uniform across all pairs $(i, j)$, we get that the result holds for 2-WISE-RANDOMSHUFFLE too!

## C.3. Results by (Yu & Li, 2023)

The crucial inequality in (Yu & Li, 2023) appears in Lemma 2.1, which is a without replacement version of the matrix Bernstein inequality. Let $X_1, \cdots, X_n$ be a set of symmetric matrices with $\bar{X} := \frac{1}{n} \sum_{i=1}^{n} X_i = 0$, and such that every $X_i$ has bounded operator norm $||X_i||_{\text{op}} \leq b$ for all $i \in [n]$. Let $\pi$ be a random permutation on $[n]$, and let $\sigma$ be a sequence of $m$ i.i.d. (with replacement) samples from $[n]$. Lemma 2.1 uses a result proved by (Hoeffding, 1963) stating that for any $s \geq 0$

$$P\left( ||\sum_{i=1}^{m} X_{\pi(i)}||_{\text{op}} \geq s \right) \leq P\left( ||\sum_{i=1}^{m} X_{\sigma(i)}||_{\text{op}} \geq s \right) \tag{9}$$

In other words, the tail probabilities of the without replacement sums are upper bounded by the tail probabilities of the with replacement sums. The latter obviously is amenable to standard Hoeffding-style bounds.

Suppose now that $\pi$ is no longer a uniformly random permutation, but one drawn from a family of $k$-wise independent distribution. Theorem 2 in (Schmidt et al., 1995) derives a Hoeffding-style bound for $k$-wise independent random variables, and in fact shows that the tail bound in the $k$-wise independent case does not differ (much) from the tail bound of the independent case! Therefore, one can expect that the high-probability first order guarantees in (Yu & Li, 2023) (albeit with slightly modified bounds) hold for K-WISE-RANDOMSHUFFLE too. We leave it as future work to understand how generating *almost k-uniform* permutations (Definition 2.2 and Algorithm 2), with error $\varepsilon$ changes the bounds derived in (Yu & Li, 2023).

