# OpenReview forum: "The benefits of full data shuffle, now with optimal I/O cost: $k$-wise independence and matrix transposition to the rescue"
_ICML.cc/2026/Conference — ICML 2026 regular_

### Official Review · Reviewer_1i3W · 2026-03-09

**Soundness:** 3
**Presentation:** 3
**Significance:** 1
**Originality:** 3
**Overall Recommendation:** 3
**Confidence:** 2

**Summary:**

This paper addresses the high I/O cost of full data shuffling between epochs in large-scale neural network training. Performing a truly random permutation incurs $\Omega(N log N / B)$ I/O operations due to arbitrary disk accesses, making it a significant bottleneck.

Authors leverage two key insight/observations to design an I/O-efficient alternative:
1. Theoretical analyses of SGD only require 2-wise independent permutations, to preserve the optimal O(1/T^2) convergence rate in smooth nonconvex optimization;
2. Real-world datasets are commonly stored in clustered (block-structured) layouts, where samples are grouped into contiguous blocks, and I/O is charged per block rather than per sample.

Building on these insights, the authors propose GEN-2-WISE-IND-PERM, an algorithm that constructs 2-wise independent permutations using a combination of matrix transposition and lightweight local shuffling. This approach ensures sufficient randomness for convergence while achieving optimal O(N/B) I/O complexity. Both theoretical experimental evaluation demonstrate the effectiveness of the proposed method.

**Compliance With Llm Reviewing Policy:**

Affirmed.

**Final Justification:**

Thank you for the thoughtful response. The additional experimental results and the clarification regarding the algorithmic design effectively address the technical concerns raised in the review. While the theoretical contribution of the paper is acknowledged, its practical relevance is likely limited, as repeated shuffling across multiple epochs appears to be diminishing in importance within contemporary and emerging deep learning training paradigms. In light of these points, I would like to raise my score to 3.

**Key Questions For Authors:**

Please see weaknesses.

**Limitations:**

The result showing that 2-wise independent permutations are sufficient for SGD convergence relies on several strong assumptions, including strong convexity, bounded domain diameter, Lipschitz continuity, smoothness, and Lipschitz-continuous Hessians. As acknowledged by the authors, prior work in this line of research typically adopts similar assumptions. However, these conditions are significantly stronger than what can reasonably be expected to hold for modern dominant model architectures, such as Transformers and diffusion models.

**Strengths And Weaknesses:**

# Strengths and weaknesses
__Strengths__:
1. The presentation is clear and easy to follow. Although the paper is theoretical, it provides detailed explanations of the fundamental concepts and assumptions, which makes it accessible even to readers with LLM engineering background.
2. The paper presents rigorous theoretical analysis. In particular, it provides formal proofs showing that the proposed shuffling method can achieve a level of randomness sufficient to satisfy the convergence assumptions of RANDOMSHUFFLE-SGD, while significantly reducing I/O complexity.

__Weaknesses__:
1. The paper is motivated by the claim that repeatedly shuffling the dataset between epochs constitutes a major I/O bottleneck during training. However, this assumption is no longer widely applicable in modern large-scale language or multimodal model training. In current practice, training often involves a single epoch over an extremely large dataset, and data shuffling can typically be performed once in advance. As a result, the practical relevance of the problem setting may be limited for contemporary training paradigms.
2. While the paper provides thorough theoretical results and claims significance in addressing real-world engineering I/O bottlenecks, the experimental evaluation is relatively limited. The experiments are conducted with N = 10007 and block size B = 14, which appears too small to convincingly demonstrate the scalability or practical effectiveness of the method in large-scale model training scenarios.
3. The proposed algorithm relies on organizing data indices into a square matrix and exploiting row- and column-wise locality to reduce I/O cost. This implicitly assumes that the dataset size can be arranged into a near-square structure and that locality is preserved within rows and columns. In large-scale datasets with highly imbalanced clusters or data sizes far from perfect squares, these assumptions may not hold, potentially causing the actual I/O complexity to deviate significantly from the theoretical analysis.

---

> ### Author Rebuttal · Authors · 2026-03-30
>
> We thank the reviewer for the review and comments. We provide below an answer to the key questions .
>
> **Question 1 (modern approaches)**
> We thank you for the expert insights. On practical relevance, our claim is not that repeated reshuffling is the dominant bottleneck in every modern LLM pipeline. Rather, the paper studies the external-memory setting where clustered storage order can harm optimization, and asks whether one can obtain the optimization benefits associated with RandomShuffle while preserving linear-scan I/O. This motivation is consistent with the discussion in the introduction, which highlights that clustered data presentation can significantly degrade training behavior.
>
> About the application to modern large languages models, the reviewer remarks that the datasets are stored and shuffled in advance.
> We wonder if our results might be of interest for dynamic settings where the large datasets evolve over time and performing regular shuffling steps might break some correlations with the temporal order in which documents are added to the dataset.
>
> Moreover, recent work continues to employ multi-epoch training [https://arxiv.org/abs/2509.22476, https://arxiv.org/abs/2412.02076v1], where the benefits of efficient reshuffling remain directly applicable.
>
> **Question 2 (experiments)**
> As the main focus of the paper is on theoretical foundations, the experiments in Section 7 should be read as a proof-of-concept rather than a large-scale systems evaluation; Section 8 already lists broader experimental validation as future work.
>
> To reinforce our analysis in Section 7, we have repeated the experiments for $N=16,000,057$ and $B=400$ over 40,000 repetitions per algorithm, which ran for 7 hours on an Intel Core Ultra 9 275HX 24 core 2.70 GHZ L3 cache 36MB, L2 cache 40 MB L1 cache 2.4 MB processor and can update the table from the paper to:
>
>
> |Algorithm | Median |      Q1--Q3 | Mean |
> |-----------------:|-----------------:|-------------------:|---------------:|
> |Gen-2-Wise-Ind-Perm    | $0$    | $0$--$0$       | $74,305$|
> |IO Shuffle (1 round)    | $60,591$   | $60,591$--$60,591$     | $60,589.5$ |
> |IO Shuffle (2 rounds)   | $79,598$   | $79,335$--$79,867$     | $79,603.5$ §|
> |CorgiPile (1 round)     | $8,017,871$ | $8,015,719$--$8,019,986$ | $8,017,852$ |
> |Fisher-Yates           | $79,600$   | $79,412$--$79,790$     | $79,600.9$ |
>
>
> Recall that in expectation for fully random shuffle is $\mathbb{E}[X]=\frac{N}{B} {B \choose 2} \cdot \frac{B-1}{N-1} = \frac{16000057}{400} {400 \choose 2} \cdot \frac{400-1}{16000057-1} = 79600.505$ supporting the claim that 2 rounds of IO Shuffle is very close to uniform in practice.
>
> **Question 3 (square matrix)**
> We would like to clarify one point that we believe may reflect a misunderstanding of the algorithmic setup. Our algorithm does not require the entire dataset to form a perfect square. \textsc{IO Shuffle} operates locally on groups of $B$ blocks at a time, viewing each such group as a $B \times B$ matrix, transposing it in fast memory, and then repeating this process. Thus, the key structural assumption is the tall-cache condition $M = \Omega(B^2)$, which ensures that these local $B^2$-element subproblems fit in memory during transposition, rather than any global square-layout assumption on the dataset.
> We will clarify this point in the final version.

---

> > ### Author Rebuttal · Reviewer_1i3W · 2026-04-05
> >
> > Thank you for the thoughtful response. The additional experimental results and the clarification regarding the algorithmic design effectively address the technical concerns raised in the review. While the theoretical contribution of the paper is acknowledged, its practical relevance is likely limited, as repeated shuffling across multiple epochs appears to be diminishing in importance within contemporary and emerging deep learning training paradigms. In light of these points, I would like to raise my score to 3.

---

> > > ### Author Response · Authors · 2026-04-05
> > >
> > > We thank the reviewer for the comments and for raising the score.

---

### Official Review · Reviewer_h3MT · 2026-03-09

**Soundness:** 4
**Presentation:** 4
**Significance:** 4
**Originality:** 4
**Overall Recommendation:** 6
**Confidence:** 4

**Summary:**

The paper studies I/O complexity of shuffling, with applications to stochastic gradient descent. Typically, stochastic gradient descent can choose a point to compute gradient on in two ways: with replacement, or without replacement. Previous works have shown that shuffling with replacement obtains convergence $O(1/T)$ while shuffling without replacement (i.e. permuting the points in each epoch) obtain faster convergence $O(1/T^2)$. However, on a machine with I/O block size $B$, shuffling with replacement requires $O(N)$ random accesses, and shuffling without replacement requires $\Omega((N/B)\log(N/B))$ I/O operations, both worse than the optimal $O(N/B)$ I/O's required to scan the input data block by block.

In light of these results, the authors consider the problem of permuting data under $k$-wise independent permutations: while the whole permutation is not uniform, fixed subsets of the indices are drawn from a marginal distribution identical (or close to in total variation distance) the uniform distribution. This is motivated by the following observation of the authors: in certain settings, the convergence guarantees of SGD only require $k$-wise independent permutations.

In this setting, the authors prove two results: whenever the cache is large (tall-cache setting) $M \geq \Omega(B^2)$, there is a $2$-wise independent shuffling algorithm with optimal $O(N/B)$ I/Os. Furthermore, the tall-cache assumption is necessary (under some additional assumptions to the algorithm). Secondly, again in the tall-cache setting, there is an approximate $k$-wise independent shuffling algorithm with $O(N/B)$ I/Os.

**Compliance With Llm Reviewing Policy:**

Affirmed.

**Key Questions For Authors:**

Algorithm 2 - why only block shuffle twice? Does repeating the procedure more times not guarantee stronger $k$-wise independence? Your lower bound rules out I/O-efficient algorithms when the cache is small. However, in the tall-cache setting, is it possible to get exact $k$-wise independence with $O(N/B)$ I/Os?

Lower Bounds - is there any hope of removing the atomic assumption? Or could we hope to improve the algorithm if we allow for non-atomic algorithms? For example, assume each input is a $q$-bit number, and somehow directly manipulate bits to obtain better algorithms without the tall-cache assumption?

**Limitations:**

yes

**Strengths And Weaknesses:**

**Strengths**

The problem of computing SGD and shuffling with low I/Os is a natural, practical problem.

The results are clean and well motivated. The algorithms are elegant and intuitive, and the paper is presented very well.

The authors establish essentially optimal results with respect to I/O complexity.

**Weaknesses**

The lower bound requires some additional assumptions (see questions below).

---

> ### Author Rebuttal · Authors · 2026-03-30
>
> We thank the reviwer for the kind and positive review and the comments. We provide below an answer to the key questions.
>
> **Question 1 (shuffle):** Algorithm 2 only shuffles twice because further rounds don't significantly increase the independence, under the "almost" independence definition. The intuition is that the first round takes elements that start inside the same block to different blocks and then round two allows elements to come back together; adding more rounds doesn't increase randomness.
>
> However, for the standard definition of independence, by using standard tools in Markov chains, it is possible to show that the distribution will converge to uniform random, if we keep applying block-shuffle and this is like how similar results have been obtained for various "card shuffling" strategies.
> We are currently unable to expand the analysis beyond what we have, even allowing for a few more rounds of shuffling. On the other hand, by looking at the lower bound arguments for the permutation problem (Agarwal and Vitter 1988) immediately yields an $\Omega(\text{sort}(k)) = \Omega(\frac{k}{B}\log_{M/B}(k/B))$ lower bound which shows that getting $\omega(N/\log_{M/B}(N/B))$-wise independence is not possible with $O(N/B)$ I/Os.
>
> **Question 2 (atomic assumption):** Proving upper/lower bounds without atomicity is notoriously difficult because algorithms could theoretically use complex arithmetic, hashing, or bit-packing to encode information.
> An interesting result appeared in [Farhadi et al, STOC 2019] that showed that, under a network coding conjecture, the I/O model of sorting without the atomicity assumption is the same as with the assumption.
> However, such result still does not completely remove assumptions since it leverages on a conjecture but we agree that it could be an interesting open problem to prove the necessity of
> tall-cache assuming a conjecture such as the network coding conjecture.

---

> > ### Author Rebuttal · Reviewer_h3MT · 2026-04-02
> >
> > I thank the authors for their answers and remain positive in my assessment.
> >
> > That being said, I'm not convinced by the answer to Question 2. If you suppose that your values are finite precision (say $q$-bits), then I don't see any fundamentally insurmountable barrier to obtaining lower bounds (e.g. the algorithm cannot do unlimited precision real-arithmetic). That being said, the lack of a non-atomic lower bound does not change my evaluation of the paper, as I find the existing work interesting already.

---

> > > ### Author Response · Authors · 2026-04-05
> > >
> > > We thank the reviewer for the useful insights and the positive review. Removing the atomicity is an interesting open question that we leave for future work. Thanks!

---

### Official Review · Reviewer_aCc7 · 2026-03-10

**Soundness:** 3
**Presentation:** 3
**Significance:** 3
**Originality:** 3
**Overall Recommendation:** 4
**Confidence:** 3

**Summary:**

The paper studies the question of limiting I/O complexity to generate random uniform permutations. An immediate motivation for this problem, given in the paper, arises from stochastic gradient descent applications. Specifically, the theoretical problem studied in the paper is this: Given N data items where each I/O refers to a block of size B of these items, find a permutation of the items aiming to be as k-wise independent as possible while minimizing the I/O.

Their main results:
- An exact 2-wise independent permutation in O(N/B) I/O
- An almost k-wise independent permutation in O(N/B) I/O

The results assume the tall cache assumption: $M = \Omega(B^2)$ where $M$ is the size of the fast memory.

They use these results for RandomShuffle convergence theoretical guarantees in strongly convex stochastic gradient descent.

**Compliance With Llm Reviewing Policy:**

Affirmed.

**Key Questions For Authors:**

- Can the authors provide an end-to-end training experiment showing the approach brings improvements in practice?

- What about non-strong convex or non-convex? Could you elaborate further on what the bottleneck is in using your techniques for more general settings?

- How common is the tall cache assumption in practice?

**Limitations:**

yes

**Strengths And Weaknesses:**

Strengths:
- A genuinely fundamental ML problem: reducing I/O cost of reshuffling is an actual important problem.
- Their results achieve optimal (linear) I/O dependency, which is neat.
- Matching lower bounds that explain well the theoretical limitations without tall-cache assumptions.


Weaknesses:
- Limitations to strongly convex, making the result not as robust as one would like.
- The experiments seem rather limited compared to what would generally be expected, though I am far from being an expert here. As indicated, they somewhat seem more like a proof of concept than a true proof of real empirical improvement.
- The writing could be further polished (typos, grammar, inconsistencies, etc).

---

> ### Author Rebuttal · Authors · 2026-03-30
>
> We thank the reviewer for the review and the comments. We provide below an answer to the key questions.
>
> **Question 1 (convexity):** That is certainly a very interesting question. First, we note that
> our algorithms ($k$-WISE RANDOMSHUFFLE) do not rely on any smoothness or convexity parameters, so they can be implemented in any setting. Second, while original works also required the component functions $f_i$ to be convex, we were already able to adapt the results that do not require this assumption.
>
> 1. For the **nonconvex** setting, the problem seems to be open even for pure RANDOMSHUFFLE. In ICML 2025, [1] states that ''In contrast, there is not much research on nonconvex cases.'' They mention that [Nguyen et al., JMLR 2021] and [Koloskova et al., arxiv 2023] indeed have some promising results using different techniques in this direction. However, they either require stronger assumptions (e.g., gradient dominance of $F$ as in [Nguyen et al.]), or are for single shuffle (and therefore exhibit worse convergence). We believe analyzing our algorithms in the nonconvex setting is a future work to be tackled **after** obtaining a cleaner picture for pure RANDOMSHUFFLE.
>
> 2. For the **non-strongly convex** case (i.e., when $f_i$s are convex but no strong-convexity on $F$), we are indeed able to adapt one result: [Theorem 3 in Mischenko et al.], also restated as Remark 1 in [Nguyen et al]. However, as mentioned in Mischenko et al., this result relies on small stepsizes and shows better convergence than SGD only for a large enough number of iterations. We are actively working in this direction, on relaxing Lipschitz smnoothness to non uniform smoothness as in [1], and on obtaining less pessimistic bounds using the primal-dual cyclic coordinate methods in [2].
>
>
> [1] He, Yu, Chen, Huang, "Revisiting Convergence: Shuffling Complexity Beyond Lipschitz Smoothness", ICML 2025.
>
> [2] Cai, Lin, Diakonikolas, "Tighter Convergence Bounds for Shuffled SGD via Primal-Dual Perspective", NeurIPS 2024.
>
>
> **Question 2 (experimental evaluation):** As the reviewer correctly noticed, results from Section 7 should be read as a proof-of-concept and we agree that an end-to-end training experimental evaluation would be
> of interest. However, the goal of our work is mainly theoretical, and we are not able to provide an end-to-end experimental analysis within the rebuttal period. We will leave this as an open question.
>
> **Question 3 (tall cache):**
> The tall-cache assumption is extremely common.
> Consider the cache-memory hierarchy. The size of the cache line for most processors is 64 or 128 bytes (e.g., Intel i5 uses 64 bytes, an SM core in Nvidia GH200 uses 128 bytes): the tall cache is satisfied when the total cache size is at least 16KB, and typically L1/L2 cache sizes by now far exceed these values (e.g., 12 MB for i5, 128KB for GH200).
> In the memory-disk hierarchy, typical block sizes are 4-16KB and the tall cache assumption is satisfied with only 256MB of memory.
>
>
> **Other comments:** We will carefully check the paper for typos, grammatical issues and inconsistencies and fix them for the final version.

---

> > ### Author Rebuttal · Reviewer_aCc7 · 2026-04-03
> >
> > Thank you for the rebuttal. I stand behind my original scoring.

---

> > > ### Author Response · Authors · 2026-04-05
> > >
> > > We thank the reviewer for the comments and the positive review.

---

### Official Review · Reviewer_rNTq · 2026-03-12

**Soundness:** 2
**Presentation:** 3
**Significance:** 2
**Originality:** 2
**Overall Recommendation:** 3
**Confidence:** 3

**Summary:**

This paper tries to solve the high I/O complexity of RANDOMSHUFFLE. A shuffling algorithm with a linear IO complexity is proposed. The algorithm can generate 2-wise independent permutations, and could be extended to k-wise independency with a small error in the probability distribution.

**Compliance With Llm Reviewing Policy:**

Affirmed.

**Key Questions For Authors:**

1. What if the question is not the strongly-convex regime?
2. What is a reasonable range for k?

**Limitations:**

yes

**Strengths And Weaknesses:**

Strengths:
1. The paper deals with an important problem in RANDOMSHUFFLE.
2. GEN-2-WISE-IND-PERM shuffles N elements using O(N/B) I/Os and produces an exact 2-wise independent permutation under the tall-cache assumption.
3. The theoretical analysis is reasonable.
4. The paper is well structured.

Weaknesses:
1. Current results are strictly theoretical, there is no experimental evaluation.
2. The analysis is only limited to the strongly-convex regime.
3. There are some typos such as “The distribution generated by Algorithm 2 generate almost k-wise independent permutatations”.

---

> ### Author Rebuttal · Authors · 2026-03-30
>
> We thank the reviewer for the review and the comments. We provide below an answer to the key questions.
>
> **Question 1 (convexity):** That is certainly a very interesting question. First, we note that
> our algorithms ($k$-WISE RANDOMSHUFFLE) do not rely on any smoothness or convexity parameters, so they can be implemented in any setting. Second, while original works also required the component functions $f_i$ to be convex, we were already able to adapt the results that do not require this assumption.
>
> 1. For the **nonconvex** setting, the problem seems to be open even for pure RANDOMSHUFFLE. In ICML 2025, [1] states that ''In contrast, there is not much research on nonconvex cases.'' They mention that [Nguyen et al., JMLR 2021] and [Koloskova et al., arxiv 2023] indeed have some promising results using different techniques in this direction. However, they either require stronger assumptions (e.g., gradient dominance of $F$ as in [Nguyen et al.]), or are for single shuffle (and therefore exhibit worse convergence). We believe analyzing our algorithms in the nonconvex setting is a future work to be tackled **after** obtaining a cleaner picture for pure RANDOMSHUFFLE.
>
> 2. For the **non-strongly convex** case (i.e., when $f_i$s are convex but no strong-convexity on $F$), we are indeed able to adapt one result: [Theorem 3 in Mischenko et al.], also restated as Remark 1 in [Nguyen et al]. However, as mentioned in Mischenko et al., this result relies on small stepsizes and shows better convergence than SGD only for a large enough number of iterations. We are actively working in this direction, on relaxing Lipschitz smnoothness to non uniform smoothness as in [1], and on obtaining less pessimistic bounds using the primal-dual cyclic coordinate methods in [2].
>
> [1] He, Yu, Chen, Huang, "Revisiting Convergence: Shuffling Complexity Beyond Lipschitz Smoothness", ICML 2025.
>
> [2] Cai, Lin, Diakonikolas, "Tighter Convergence Bounds for Shuffled SGD via Primal-Dual Perspective", NeurIPS 2024.
>
>
> **Question 2 (range of $k$):** Regarding reasonable ranges for $k$, our second algorithm produces approximate $\sqrt{N/B}$-wise independence which one can consider to be a high level of independence.
> For instance: simple hashing strategy of linear probing requires 5-wise independent hash functions; Cuckoo Hashing $O(\log n)$-wise independent hash functions are known to be sufficient; sketches, like CountSketch, require 4-wise independence.
>
> **Other comments:** We will carefully check the paper and fix typos for the final version. As the reviewer noticed, the main focus of this paper is mainly theoretical. The experimental evaluation is clearly relevant and deserves future work; however, we would like to remark that we provide novel and foundational results and that the ICML call for papers allows for theoretical papers.

---

### Decision · Program_Chairs · 2026-04-30

**Decision:**

Accept (regular)

**Comment:**

One reviewer has very strong support, while the other reviewers have more tempered enthusiasm. The strong support is based on this being a fundamental and important theoretical achievement. The negative support asks for more experiments, extensions to the non-convex setting (which is not even solved for pure RandomShuffle), and claims that modern LLM training are moving towards single-epoch training making the paper's approach less relevant.

I think I'm inclined to go with the strong supporter here, as this is an important theoretical contribution to the community. It is framed as a theoretical paper, it should not have to solve the non-convex setting for pure RandomShuffle itself, and the engineering decisions of moder LLM training may be transient.